# New Opportunities for Preoperative Diagnosis of Medullary Thyroid Carcinoma

**DOI:** 10.3390/biomedicines11051473

**Published:** 2023-05-18

**Authors:** Sergei A. Lukyanov, Sergei V. Sergiyko, Sergei E. Titov, Dmitry G. Beltsevich, Yulia A. Veryaskina, Vladimir E. Vanushko, Liliya S. Urusova, Alexander A. Mikheenkov, Evgeniya S. Kozorezova, Sergey L. Vorobyov, Ilya V. Sleptsov

**Affiliations:** 1Department of General Surgery, South Ural State Medical University, 454092 Chelyabinsk, Russia; 111lll@mail.ru (S.A.L.); ssv_1964@mail.ru (S.V.S.); 2Laboratory of Molecular Genetics, Department of the Structure and Function of Chromosomes, Institute of Molecular and Cellular Biology, SB RAS, 630090 Novosibirsk, Russia; 3AO Vector-Best, 630117 Novosibirsk, Russia; 4Department of Natural Sciences, Novosibirsk State University, 630090 Novosibirsk, Russia; 5Department of Surgery, National Medical Research Center for Endocrinology, 115478 Moscow, Russia; belts67@gmail.com (D.G.B.); vanushko@gmail.com (V.E.V.); liselivanova89@yandex.ru (L.S.U.); mikheenkov_alexander@mail.ru (A.A.M.); 6Laboratory of Gene Engineering, Institute of Cytology and Genetics, SB RAS, 630090 Novosibirsk, Russia; microrna@inbox.ru; 7Department of Cytopathology, National Center for Clinical Morphological Diagnostics, 192071 Saint Petersburg, Russia; kozorezovaes@yandex.ru (E.S.K.); slvorob@gmail.com (S.L.V.); 8Department of Endocrinology and Endocrine Surgery, Saint Petersburg State University N.I. Pirogov Clinic of High Medical Technologies, 190121 Saint Petersburg, Russia; newsurgery@yandex.ru

**Keywords:** medullary thyroid carcinoma, fine-needle aspiration samples, molecular testing, miRNA-375, calcitonin

## Abstract

The preoperative diagnostics of medullary thyroid carcinoma (MTC), including the measuring of the blood calcitonin level, has a number of limitations. Particular focus has recently been placed on the role of miRNAs in the development of various malignant tumors; a comparative analysis of accuracy of the existing methods for MTC diagnosis with a novel diagnosis method, evaluation of the miRNA-375 expression level, was performed in this study. The expression level of miRNA-375 in cytology samples from 555 patients with the known histological diagnosis, including 41 patients with confirmed postoperative diagnosis of MTC, was assessed. The diagnostic parameters of the basal calcitonin level, calcitonin in wash-out fluid from the FNAB needle, and miRNA-375 were compared. An assessment of the miRNA-375 expression level made it possible to detect all the MTC samples with a 100% accuracy among all the 555 cytology specimens, as well as in non-informative FNAB specimens, and specimens from the ipsilateral thyroid lobe. Parameters such as sensitivity, specificity, PPV, and NPV were 100%. The miRNA-375 level, unlike calcitonin, does not correlate with tumor volume, so it does not have the so-called “gray zone”. An assessment of the miRNA-375 expression allows one to accurately distinguish MTC from other malignant and benign thyroid tumors.

## 1. Introduction

The incidence rate of medullary thyroid carcinoma (MTC) is ~5% among all the thyroid cancer cases [1]. However, this parameter has noticeably decreased over the past three decades to reach 1–2% because of the substantially improved detectability of papillary thyroid carcinoma [2]. Despite its low incidence rate, MTC is the cause of death for 13% of all thyroid cancer patients [3]. MTC has an aggressive clinical course; the 5- and 10-year survival rates are 65% and 50%, respectively [4]. Early detection of MTC significantly increases the overall survival in patients, while delayed diagnosis and/or incomplete surgical treatment correlate with a less favorable prognosis [5]. Furthermore, medullary thyroid carcinoma is a component of multiple endocrine neoplasia type 2 syndrome (MEN2A and MEN2B) in 25% of cases, suggesting that the preoperative assessment of mutations in the RET proto-oncogene, and screening for primary hyperparathyroidism and pheochromocytoma should be performed, and the minimal surgery extent should involve a total thyroidectomy and prophylactic central neck dissection [6]. Therefore, preoperative detection plays a decisive role in improving the survival rate in patients with MTC.

Measuring the level of calcitonin secreted by thyroid C cells is a high-sensitivity method for the preoperative diagnosis of MTC. However, routine measurements of basal calcitonin levels can be interpreted ambiguously in the high-normal range: hypercalcitoninemia can be caused by other conditions such as thyroiditis, sepsis, hypercalcemia, hypergastrinemia, neuroendocrine tumors, chronic kidney disease, chronic obstructive pulmonary disease, and pseudohypoparathyroidism [7,8,9], and also by pharmacological agents (i.e., proton-pump inhibitors, glucocorticoids, and β-blockers) [10]. For this reason, there is no consensus among clinicians regarding whether it is reasonable to measure serum calcitonin levels in all patents with thyroid nodules. The authors of American clinical practice guidelines provide neither pros nor cons for this laboratory test, thus offering the clinicians an opportunity to independently solve their question in a particular medical center, whereas European researchers insist on measuring routine calcitonin level in patients with thyroid nodules [6,11,12,13].

At calcitonin levels above 100 pg/mL, the diagnosis of MTC is beyond doubt. Diagnostic uncertainty exists at low calcitonin levels (10–100 pg/mL) and is known as the so-called “gray zone”. In these cases, it is recommended to use the calcium stimulation test to increase sensitivity to basal serum calcitonin [14,15]. Meanwhile, recent studies have demonstrated that measurements of stimulated calcitonin levels do not noticeably increase the accuracy of the preoperative detection of MTC in the “gray zone”, since the stimulation tests have no reference values and do not differ for patients with neoplastic and secondary C-cell hyperplasia in terms of calcitonin range amplitude [16]. It makes sense to perform the calcitonin stimulation test only to rule out extrathyroidal calcitonin-secreting tumors with no C-cell calcitonin-stimulating receptors.

Taking into account that approximately 50% of all patients with MTC will preoperatively have indeterminate cytology results (Bethesda categories III, IV, and V), while the basal calcitonin level may lie in the “gray zone” or remain undetected, there will inevitably be cases of surgery refusal or choosing an inadequate surgery extent [17,18].

Molecular testing has recently been increasingly used to improve diagnosis making and optimize the treatment of patients with thyroid nodules for whom indeterminate cytology results have been obtained [19]. These tests can substantially improve the quality of the preoperative detection of various types of thyroid cancer. Thus, the Afirma RNA-sequencing MTC classifier, where expression of 108 genes in the wash-out fluid from fine-needle aspiration of thyroid nodules is analyzed, showed 100% sensitivity and specificity in MTC detection in 21 out of 211 patients cytologically classified as belonging to Bethesda III–VI categories [20]. Another test, ThyraMIR, based on the analysis of the expression of ten different miRNAs also made it possible to detect MTC in 11 out of 314 patients cytologically classified as belonging to Bethesda III–IV categories with 100% sensitivity and specificity [21]. Meanwhile, only miRNA-375 was the marker enabling 100% accuracy in detecting MTC.

The overexpression of miRNA-375 in MTC patients was first reported in 2011 by Abraham et al., who compared the expression levels of this miRNA between the sporadic and inherited MTC, but no comparison with other types of thyroid cancer was carried out [22]. Titov et al. (2016) were the first ones to suggest using mRNA-375 as a molecular test component in the diagnosis of MTC [23]. In 2018, Romeo et al. also pointed at the important role of miRNA-375 as a marker of the progression and aggressive course of MTC [24]. However, in this study, the level of miRNA-375 was assessed in the postoperative period in the patient’s plasma.

Therefore, the aim of the present study is to compare the accuracy of the MTC detection using the analysis of the miR-375 expression in the cytological material obtained by FNAB and the determination of the basal calcitonin level in these patients.

## 2. Materials and Methods

### 2.1. Clinical Material

A total of 555 patients were enrolled in the study. The data on 525 patients (group 1 of patients) were taken from our previous studies performed to evaluate the diagnostic accuracy of the Thyroid-INFO test [25,26,27]. In this group, in previous studies, the expression level of miRNA-375 was determined for all the samples, and, for some samples (n = 354), the basal calcitonin level was known. In these studies, the criteria for inclusion in the study were as follows: patients’ age being at least 18 years, the presence of primary cytological slides with a sufficient amount of cellular material (Bethesda category II–VI), and postoperative histological conclusion being known. Expression of miRNA-375 was determined in the material washed off from cytological slides (Figure 1). The distribution of pathologies was as follows: goiters, 79 (15%) cases; follicular adenoma (FA), 219 (41.2%); oncocytic adenoma (OA), 60 (11.4%); follicular thyroid carcinoma (FTC), 30 (5.7%); oncocytic thyroid carcinoma (OTC), 13 (2.4%); papillary thyroid carcinoma (PTC), 108 (20.6%); medullary thyroid carcinoma (MTC), 11 (2.1%); and parathyroid adenoma (PTA), 5 (0.9%) cases. The proportion of malignant tumors was 30.8%. There were 83 men (15.8% of cases), mean age 53 (18–83) years; and 442 women (84.2%), mean age 55 (18–86) years.

Since the percentage of MTC in group 1 patients is low, cytology specimens obtained by FNAB were additionally examined in 30 patients with MTC (control group 2) who were receiving treatment at the National Medical Research Center for Endocrinology. The final diagnosis for this group of patients, as with for group 1, was made using the postoperative histological report. Histological examination was carried out by pathologists from the corresponding institution. The miRNA-375 level was evaluated in the material washed off from cytological slides in all the group 2 patients (Figure 1); in one patient, it was also assessed in the ipsilateral thyroid lobe; and in eight patients, in the contralateral lobe. In two cases, the material collected from the nodules was found to be cytologically non-informative (Bethesda I); this material was also attributed to the ipsilateral thyroid lobe.

The study was approved by the Ethics Committee of the Southern Ural State Medical University (Protocol No. 3). The material was collected in compliance with the current legislation of the Russian Federation; each patient had provided written informed consent; all the data were depersonalized.

### 2.2. Molecular Analysis

A retrospective analysis of miRNA-375 expression level measured using comprehensive molecular testing (Thyroid-INFO) was carried out for group 1 patients. The test involved an analysis of expression of the *GCM2* and *HMGA2* genes, mRNA-146b, -221, and -375, the ratio of mitochondrial to nuclear DNA, and identification of the BRAF V600E mutation [27]. Expression of miRNA-375 was normalized to the geometric mean level of miRNA-29b, -23a, and -197.

In the specimens from group 2 patients, the miRNA-375 levels were measured under this study. The oligonucleotides for detecting miRNA-375 were as follows: TaqMan probe (R6G)-ATTCGCACC(T-BHQ1)CGACACGACTCACGCGA, forward primer ACAGCTTTGTTCGTTCGGC, reverse primer CTGAGGCTCACTGAGACCT, and long primer for reverse transcription GTCGTGTCTGAGGCTCACTGAGACCTATTCGCACCTCGACACGACTCACGCGA.

The oligonucleotides for detecting miRNA-29b were as follows: TaqMan probe (R6G)-TTCGCACCC(T-BHQ1)CGACACGACAACACTGAT, forward primer CAGCACTAGCACCATTTGAA, reverse primer CTGAGGCTCACTGAGACCT, and long primer for reverse transcription GTCGTGTCTGAGGCTCACTGAGACCTTTCGCACCCTCGACACGACAA(C-LNA)AC(T-LNA)GAT. 

The oligonucleotides for detecting miRNA-23a were as follows: TaqMan probe (R6G)-TTCGCACCC(T-BHQ1)CGACACGACGGAAATCC, forward primer CAGCACATCACATTGCCAG, reverse primer CTGAGGCTCACTGAGACCT, and long primer for reverse transcription GTCGTGTCTGAGGCTCACTGAGACCTTTCGCACCCTCGACACGACGGAAATCC.

The oligonucleotides for detecting miRNA-197 were as follows: TaqMan probe (R6G)-TTACGCACC(T-BHQ1)GCCACGACGCTGGGTG, forward primer CCACGTTCACCACCTTCTC, reverse primer GTGAAGCAGACAGACACAA, and long primer for reverse transcription GTCGTGGGTGAAGCAGACAGACACAATTACGCACCTGCCACGACGCTGGGTG.

Nucleic acids were isolated from cytological slides using the procedure reported previously with a certain modification, namely, each dried cytological preparation was washed into a tube using three portions of guanidine lysis buffer (200 µL each) [23]. Next, the specimen was mixed and incubated in a thermoshaker at 60 °C for 15 min. An equal volume of isopropanol was then added; the tube contents were thoroughly mixed and left at room temperature for 5 min. After centrifuging for 10 min at 14,000× *g*, the supernatant was removed, and the precipitate was washed with 500 µL of 70% ethanol and 300 µL of acetone. Finally, RNA was dissolved in 200 µL of deionized water. If not analyzed immediately, RNA samples were stored at −20 °C for future use.

miRNA was detected by stem-loop real-time PCR [28]. The reverse transcription reaction was carried out individually for each miRNA, followed by real-time PCR, according to the procedure described previously [23]. A single analysis run was performed for each sample. The miRNA-375 levels were normalized to the geometric mean of the levels of three reference miRNAs (-197-3p, -23a-3p, and -29b-3p) using the 2^-ΔCq^ method [29]. The cut-off for MTC used in this study was 2.8; it was determined in previous studies using the C4.5 decision tree algorithm [25].

### 2.3. Calcitonin Level Measurements

Basal plasma calcitonin levels were measured in 354 patients in group 1 and all the group 2 patients; the calcitonin level in the wash-out fluid from fine-needle aspiration was evaluated in 10 patients. The measurements were performed by solid-phase chemiluminescent enzyme immunoassay. Calcitonin levels were determined preoperatively at medical centers where the patients were receiving treatment. The cut-off values for calcitonin (regardless of sex) were as follows: <10 pg/mL—normal value; 10–100 pg/mL—the “gray zone”; and >100 pg/mL—medullary thyroid cancer.

### 2.4. Statistical Analysis

The data were analyzed using the Excel (Microsoft, Redmond, WA, USA) or Statistica 13.3 software (TIBCO Software, Palo Alto, CA, USA). The diagnostic characteristics were determined using the standard 2 × 2 contingency tables comparing the qualitative results of binary molecular tests (positive or negative) with respect to the reference diagnoses made by histopathological examination. The confidence intervals for sensitivity, specificity, and accuracy were calculated using the Clopper–Pearson method, the confidence intervals for the predictive values are the logit confidence intervals. The correlation between tumor size and calcitonin level was assessed using the Spearman’s rank correlation coefficient. Two independent samples were compared by quantitative traits via the Mann–Whitney U test.

## 3. Results

The miRNA-375 expression levels in preoperative cytology specimens collected from MTC patients were assessed for different types of thyroid and parathyroid tumors (Figure 2 and Table 1; see also Appendix A). In the group of patients with MTC, two cytology specimens (Bethesda I, non-informative) were excluded from analysis.

In patients with MTC, the miRNA-375 level differed statistically significantly from those for all other types of tumors (*p* = 0.000001). Although its expression level was also increased in patients with PTC and PTA, the range of variation was significantly higher in the case of MTC. The ratio between the mean miRNA-375 level in patients with MTC and PTC was 76.4.

The diagnostic characteristics of the basal calcitonin level were evaluated in 384 patients in groups 1 and 2. In 50 of those, calcitonin levels increased above 10 pg/mL were observed (in 40 patients with MTC and 10 patients with other thyroid tumors). In one patient with medullary thyroid carcinoma, the calcitonin level was <1.0, while the miRNA-375 level was 3.35. The miRNA-375 expression level increased above the assumed cut-off value (2.8) was observed in all the patients with MTC in groups 1 and 2, while remaining low in patients with other pathologies (Table 2).

In 10 (24.4%) out of 41 patients with MTC (in groups 1 and 2 pooled into a single array), the basal calcitonin level lay within the “gray zone” from 10 to 100 pg/mL; 51.9 (24.6–59.9); minimum, 13 pg/mL; maximum, 98 pg/mL. Assessment of the calcitonin level in the wash-out fluid from fine-needle aspiration from the thyroid nodules was informative in 10 (100%) cases and revealed that it was significantly increased (>2000 pg/mL). The median expression level of miRNA-375 in the same 10 patients was 8.9 (7.9–11.7).

The minimal volume of tumor tissue (according to postoperative histological examination) and minimal preoperative blood calcitonin level in group 2 patients with MTC was 0.03 cm^3^ (tumor diameter, 4 mm) and 13 pg/mL, respectively; the minimal expression level of miRNA-375 was 3.35. The maximum volume was 33 cm^3^ (tumor diameter, 4 cm); calcitonin level was >10,000 pg/mL (the hook effect); and miRNA-375 level was 31.8. An analysis of correlation between tumor volume in MTC patients and the levels of calcitonin and miRNA-375 was carried out; the correlation between calcitonin or miRNA-375 levels and patient’s age were also determined for comparison (Table 3).

There is a strong positive correlation between tumor size and calcitonin level: the larger the tumor volume, the higher the basal blood calcitonin level. The low Spearman’s rank correlation coefficient between the tumor size and miRNA level in cytology specimens indicates that these characteristics are not correlated in MTC patients. There is also no correlation between age and basal calcitonin or miRNA-375 level.

In order to assess the differences in miRNA-375 contents in different regions of the thyroid gland, its level was measured directly in the tumor nodule, as well as in the ipsilateral and contralateral thyroid lobes in some group 2 patients (Table 4).

The miRNA-375 level was significantly increased in all the cytology specimens collected from the tumor node (ME = 14.9 (8.5–20.4)). In six out of eight patients, the miRNA-375 level in the contralateral thyroid lobe was extremely low (ME = 0.035 (0.02–0.04)). Expression of miRNA-375 was also increased in two patients in the contralateral thyroid lobe and in all three cases in the ipsilateral thyroid lobe (ME = 7.9 (5.6–11.5)). In the two cases when miRNA-375 was overexpressed in the contralateral thyroid lobe, a postoperative histological examination revealed multifocal tumor growth sites.

## 4. Discussion

We have comparatively analyzed the accuracy of measuring the basal calcitonin level and miRNA-375 expression for the preoperative detection of MTC. According to our findings, the specificity and sensitivity of calcitonin level measurements was 97.1% and 97.6%, respectively, which is consistent with the literature data. In the study based on an analysis of calcitonin level in 72,368 patients with thyroid nodules, Verbeek et al. demonstrated that for the calcitonin threshold value of 10 pg/mL, the sensitivity of the test is 100% (95% CI: 99.7–100); and the specificity is 97.2% (95% CI: 95.9–98.6) [30]. For the combined test to measure the basal and stimulated calcitonin levels, the sensitivity ranged from 82 to 100%, while the specificity ranged from 99 to 100%. Hence, the calcitonin test almost completely prevents the risk of missing a diagnosis of medullary thyroid carcinoma, wherein the positive predictive value for the increased calcitonin level is as low as 7.7%; i.e., more than 90% of patients with an increased calcitonin level did not actually have MTC, and the surgery was either too radical or was not needed at all.

In our study, for a sample where MTC was 10% of all samples, the PPV was 80%. However, although sensitivity and specificity characterize a test independently of the incidence of cancer, NPV and PPV depend on the incidence. Assuming a fixed sensitivity and specificity, Bayes’ theorem can predict the dependence of a test’s NPV and PPV on the incidence of cancer, so with the incidence of MTC among all thyroid nodules (0.25%), the PPV will also be 7.7% (95% CI: 4.1–13.4%).

The assessment of the miRNA-375 expression level was devoid of these drawbacks. According to our data, this preoperative marker made it possible to accurately distinguish MTC from other malignant and benign tumors of the thyroid and parathyroid glands. The difference in the mean expression levels was >70-fold for MTC vs. PTC and >1600-fold for benign tumors and goiters. Parameters such as sensitivity, specificity, PPV, and NPV were 100%.

Ultrasound-guided FNAB is a reliable diagnostic tool for the primary assessment of thyroid nodules; however, it has a number of shortcomings in the MTC cases. During preoperative FNAB, the diagnosis of MTC is made only in 50% of patients; cancer is suspected in 20% of them, and the FNAB results are indeterminate in 30% of patients [18,31]. Boi et al. were the first to recommend measuring calcitonin levels in the wash-out fluid from the fine-needle aspiration of thyroid nodules to enhance the diagnostic quality in patients with MTC [32]. However, this procedure of the preoperative diagnosis of MTC has a number of limitations, primarily due to its complexity and availability [33]. In our study, a significant increase in calcitonin levels in the wash-out fluid from the fine-needle aspiration of thyroid nodules was detected in all ten patients with MTC with the calcitonin level lying in the “gray zone”, thus verifying the diagnosis of MTC. However, the biopsy had to be repeated and the medical staff member performing it had to be extremely skilled, since the nodule diameter ranged from 0.4 to 0.9. The use of miRNA-375 also ensured a 100% accuracy in diagnosis making by analyzing the primary stained cytology preparation. We believe that the accuracy of miRNA-375 as a marker of MTC does not correlate with tumor size, in contrast to calcitonin levels, where it increases as tumor volume increases, but may also increase for other reasons.

According to the findings reported in different publications, approximately 50% of cases of elevated calcitonin levels (20–100 pg/mL) are associated with secondary C-cell hyperplasia [34,35], which also occurs in up to 30% of various (non-MTC) thyroid nodules [36,37,38]. Based on these literature data, it is fair to assume in our situation that patients without MTC could have C-cell hyperplasia (from 25 to 150 cases). Nevertheless, an elevated miRNA-375 expression level was observed in none of them, since no false positive results were detected. This can indicate indirectly that C-cell hyperplasia does not affect the miRNA-375 level.

An unexpected feature of our study was that the miRNA-375 level was also very high in the ipsilateral thyroid lobe, while being low in the contralateral thyroid lobe, provided that no multifocal tumor growth took place. It can be a novel additional tool for dynamic follow-up if patient’s remaining thyroid lobe was not removed for some reason.

The apparent advantages of using the miRNA-375 expression level for the preoperative diagnosis of MTC are as follows: (1) it can be easily measured (it is the conventional real-type PCR test kit, so this test can be performed in almost any clinical laboratory); (2) this marker can be detected beyond a comprehensive molecular test, so the analysis cost can be significantly reduced; and (3) the specimen sampling procedure is rather simple: the test “works” even for the non-informative (few-celled) FNAB specimens, which is extremely important for medullary thyroid microcarcinomas, where the physician performing a biopsy needs to be extremely skilled. This study can be carried out on stained archival cytological preparations. Since measuring the calcitonin level is more common, miRNA-375 can be used for diagnosis verification when the calcitonin level falls in the “gray zone” without the need to make a new biopsy.

The disadvantages of the presented study include its retrospective nature and the relatively small number of cases of MTC.

## 5. Conclusions

Our results show that the assessment of miRNA-375 expression enables the high-accuracy detection of medullary thyroid carcinoma, but additional studies need to be performed because of the relatively small sample of patients.

## Figures and Tables

**Figure 1 biomedicines-11-01473-f001:**
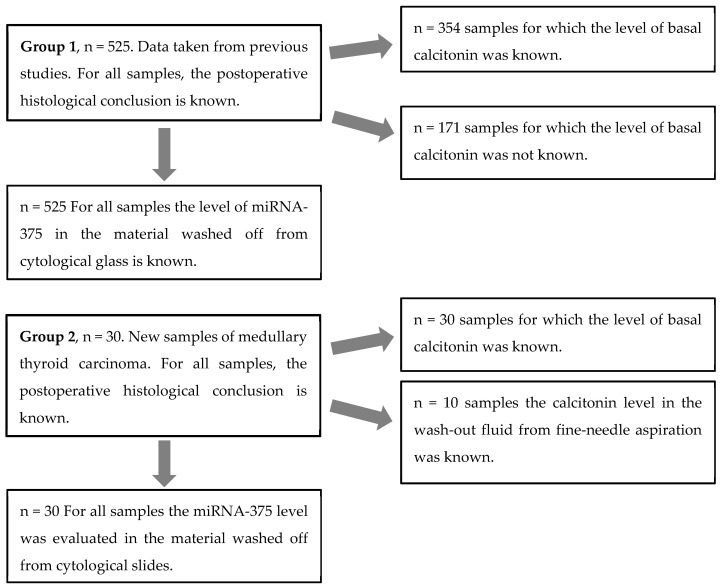
Scheme for analyzing miRNA-375 and preoperative basal calcitonin in group 1 and 2 patients.

**Figure 2 biomedicines-11-01473-f002:**
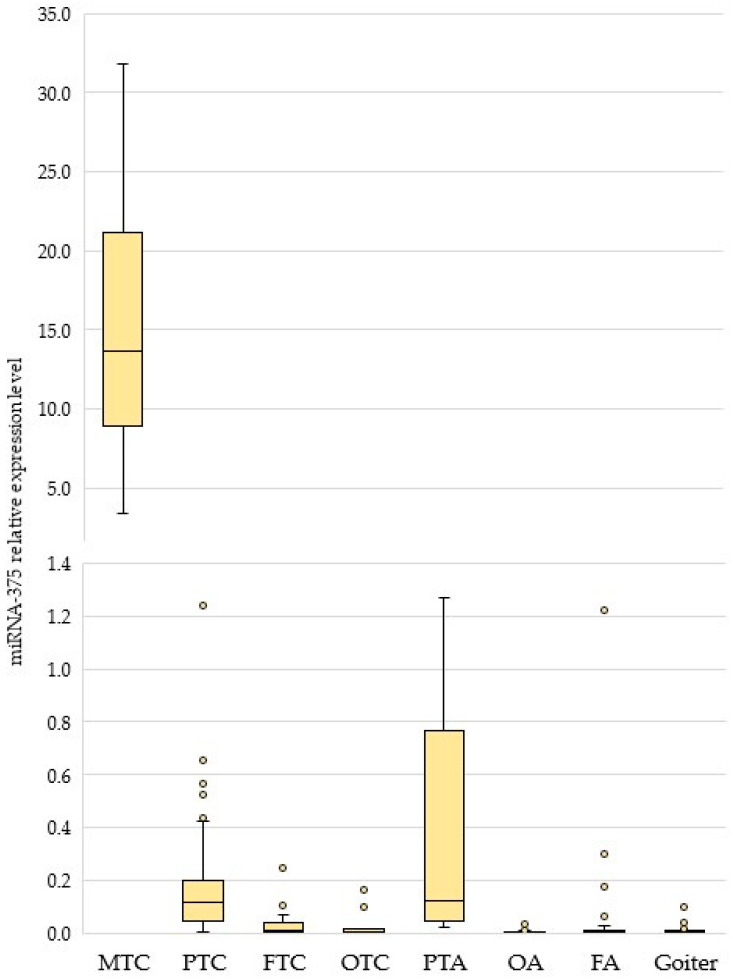
The relative expression level of miRNA-375 in goiters and thyroid/parathyroid tumors. The figure presents the median value, upper and lower quartiles, and non-outlier range, and outliers (circles). MTC, medullary thyroid carcinoma; PTC, papillary thyroid carcinoma; FTC, follicular thyroid carcinoma; OTC, oncocytic thyroid carcinoma; PTA, parathyroid adenoma; OA, oncocytic adenoma; and FA, follicular adenoma.

**Table 1 biomedicines-11-01473-t001:** Levels of microRNA-375 expression in different tumor types.

	MTC (n = 39)	PTC (n = 108)	FTC (n = 30)	OTC (n = 13)	PTA (n = 5)	OA (n = 60)	FA (n = 219)	Goiter (n = 79)
ME	14.1	0.113	0.011	0.004	0.124	0.003	0.004	0.004
Q1	9.1	0.043	0.003	0.002	0.064	0.001	0.001	0.002
Q3	21.2	0.197	0.034	0.009	0.262	0.004	0.012	0.007
M	16.1	0.155	0.028	0.024	0.349	0.004	0.019	0.008

ME, median value; Q1, Q3, the first and third quartiles; M, mean value. MTC, medullary thyroid carcinoma; PTC, papillary thyroid carcinoma; FTC, follicular thyroid carcinoma; OTC, oncocytic thyroid carcinoma; PTA, parathyroid adenoma; OA, oncocytic adenoma; and FA, follicular adenoma.

**Table 2 biomedicines-11-01473-t002:** The diagnostic characteristics of basal calcitonin and miRNA-375 levels for MTC detection (including the 95% confidence interval).

Result	miRNA-375 Level (n = 555)	Calcitonin Level (n = 384)
False positive	0	10
False negative	0	1
True positive	41	40
True negative	514	333
Specificity, %	100 (99.3–100)	97.1 (94.7–98.6)
Sensitivity, %	100 (91.4–100)	97.6 (87.1–99.9)
PPV, %	100	80 (62.3–89.9)
NPV, %	100	99.7 (98.3–99.9)

PPV, positive predicted value; NPV, negative predicted value.

**Table 3 biomedicines-11-01473-t003:** The correlation between tumor volume and calcitonin level or microRNA-375 expression level in MTC patients (Spearman’s coefficient, including the 95% confidence interval).

Parameter	Spearman’s Coefficient	*p*-Value
Tumor volume/calcitonin	0.85 (0.66–0.97)	0.00001 *
Age/calcitonin	0.19 (0.14–0.51)	0.17
Tumor volume/miRNA-375	0.31 (−0.06–0.65)	0.093
Age/miRNA-375	0.23 (0.11–0.57)	0.22

* significant correlation.

**Table 4 biomedicines-11-01473-t004:** miRNA-375 expression levels in different regions of the thyroid gland in patients with MTC.

No.	TNM Stage	Level of miRNA-375 from the Nodule	Level of Mirna-375 from the Contralateral Thyroid Lobe	Level of miRNA-375 from the Ipsilateral Thyroid Lobe
1	T2N1bM0	17.4	0.04	Not performed
2	T2N0M0	11.5	0.04	Not performed
3	T1bN1bM0	6.7	0.01	Not performed
4	T1bN1aM0	17.3	0.03	Not performed
5	T1bN0M0	7.1	0.05	Not performed
6	T1a(m)N1aM0	FNAB is non-informative	5.6	10.2
7	T1aN0M0	21.4	0.02	Not performed
8	T1b(m)N1bM1	25.5	15.4	Not performed
9	T1aN0M0	12.6	Not performed	18
10	T1bN0M0	FNAB is non-informative	Not performed	5.4

## Data Availability

The data are contained within the article or Appendix A.

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
