# Peer review of "New Opportunities for Preoperative Diagnosis of Medullary Thyroid Carcinoma"

_biomedicines, 2023, doi:10.3390/biomedicines11051473_

Round 1
Reviewer 1 Report
he preoperative diagnosis of medullary thyroid carcinoma (MTC) through measuring blood calcitonin levels has limitations. Recent studies have focused on the role of miRNAs in the development of malignant tumors. This study compared the accuracy of existing MTC diagnosis methods with a novel method that evaluates miRNA-375 expression levels in cytology samples from 555 patients, including 41 with confirmed postoperative diagnosis of MTC. The results showed that miRNA-375 expression level accurately detected all MTC samples with 100% accuracy and distinguished MTC from other malignant and benign thyroid tumors, outperforming other molecular tests and calcitonin measurements.
Looking into miR-375 in particular was reported before and even used in genetic testings of commercial companies. However, one of the strength of the study is measuring the expression level in various types of thyroid diseases, using multiple normalization reference, adding 2 other known miRNAs (146 and 221).
*It is unclear PCR expression level calculation was compared to what in this formula: 2^-dCq Line 160. Which group was compared to? and which endogenous control (miRNA) was used?
*Another inquiry, the cutoff value line 206 is log transformed or raw value?
*What is the value of Table 3 Lines 223-230?
*The study only evaluated the accuracy of miRNA-375 expression level and basal calcitonin level measurements (the status quo) in preoperative diagnosis of MTC, without considering the combined use of both markers.
*The study also did not investigate the correlation between miRNA-375 expression level and the effect of other factors, such as patient age or gender, on the accuracy of the marker.Regression analysis is suggested.
Author Response
*It is unclear PCR expression level calculation was compared to what in this formula: 2^-dCq Line 160. Which group was compared to? and which endogenous control (miRNA) was used?
Formula 2^(-dCq) implies that we count the relative miRNA expression (with respect to a certain reference value) rather than absolute one. In this work (as well as our other studies), we use the geometric mean Cq value of three miRNAs (-197-3p, -23a-3p, and -29b-3p) as a reference. As opposed to formula 2^(-ddCq), in this formula the tumor data are not normalized with respect to some normal tissue.
*Another inquiry, the cutoff value line 206 is log transformed or raw value?
No, it is not a logarithmic scale. The value 2.8 means that expression of miRNA-375 was higher than the expression level of three reference miRNAs 2.8-fold: 2^-(Cq(miR-375) – geo mean (Cq(miR-197 + 23a + 29b))) = 2.8.
*What is the value of Table 3 Lines 223-230?
The correlation between tumor size and diagnostic markers of thyroid cancer is very important. At the preoperative stage when the tumor is small, the low calcitonin level may cause a missed diagnosis of MTC, or the calcitonin level will fall in the "gray zone" (< 100 pg/mL), which reduces its diagnostic value. Our study has demonstrated that assessment of the miRNA-375 level is devoid of this drawback and shows excellent result when used for both microcarcinoma and large tumors.
*The study only evaluated the accuracy of miRNA-375 expression level and basal calcitonin level measurements (the status quo) in preoperative diagnosis of MTC, without considering the combined use of both markers.
There is no certainty regarding discussion of this topic, since measuring the calcitonin level is a well-mastered procedure, but measuring the miRNA-375 level proposed in our study was much more advantageous. Nevertheless, we have added the following sentence to the Discussion section:
Since measuring the calcitonin level is more common, miRNA-375 can be used for diagnosis verification when the calcitonin level falls in the “gray zone” without the need to make a new biopsy.
*The study also did not investigate the correlation between miRNA-375 expression level and the effect of other factors, such as patient age or gender, on the accuracy of the marker. Regression analysis is suggested.
We did not study the correlation between miRNA-375 and other factors, since the accuracy of medullary cancer detection using it was 100%, suggesting no influence of any other factors. We studied its correlation with tumor size because it is known for calcitonin, and we wanted to explicitly demonstrate that there is no such correlation for miRNA-375. Nonetheless, we added the data on correlation between calcitonin/miRNA-375 and patients' age to Table 3 (no correlation was detected).
Reviewer 2 Report
Excellent paper.
In the result and discussion, the author should clearly describe
1) Familial or sporadic type of MTC.
2) Any C cell hyperplasia in this study?
And discuss the possible influence
Author Response
In the result and discussion, the author should clearly describe
1) Familial or sporadic type of MTC.
The presence of mutations in the RET gene (exons 10, 11, 13, 14, and 15) was tested in 23 out of 41 patients with MTC. Mutations were detected in four patients: (exon 11) С634T in heterozygous state, (exon 11) С634G in heterozygous state, (exon 14) V804M in heterozygous state, and (exon 11) C634F in heterozygous state. Since the data on mutations are incomplete, we decided not to report them in the manuscript.
2) Any C cell hyperplasia in this study?
We have not been tested C-cell hyperplasia in the samples, while it might be present as judged from the number of samples used in this study. It is known that it may affect the calcitonin level, but since analysis of the miRNA-375 expression level yielded no false positive results, C-cell hyperplasia does not influence this marker.
We have added a comment about this fact to the Discussion section:
According to the findings reported in different publications, approximately 50% of cases of elevated calcitonin levels (20–100 pg/mL) are associated with secondary C-cell hyperplasia [34, 35], which also occurs in up to 30 cases of different goiter (non-MTC) cases [36, 37, 38]. Based on these literature data, it is fair to assume in our situation that patients without MTC could have C-cell hyperplasia (from 25 to 150 cases). Nevertheless, elevated miRNA-375 expression level was observed in none of them, since no false positive results were detected. This can indicate indirectly that C-cell hyperplasia does not affect the miRNA-375 level.
And discuss the possible influence
This comment needs to be clarified.
Reviewer 3 Report
The present study evaluates the expression of microRNA 375 level in cases with known CMT.
2.1 CLINICAL MATERIAL
The definition of group 1 – 11 cases with known CMT – confirmed by pathology report (after surgical removal) – retrospective analysis – it is not clear in pathology reports of FNA reports were analysed. It is indirectly suggested that from the total of 525 cases, 375 had FNAB specimens, which were analysed for the microRNA mutation, 11/375 being with definitive CMT diagnostic.
The definition of group 2 – 30 cases of CMT patients receiving treatment at the .... what do you mean under this statement? How was the definitive diagnostic for CMT made – FNAB only or surgery – if it was surgery why are these cases not comprised in the first group? What is different in group 2 versus group1 ... if they have no final pathology report confirming CMT 0 what was the golden standard for CMT diagnostic ...
This paragraph needs clarification.
2.2 MOLECULAR ANALYSIS
Group1 - mutation were part of a more comprehensive molecular testing.
Group 2 – define what you understand under” de novo” – did you test miRNA 375 expression only?
2.3 Calcitonin
Defined values of Calcitonin, threshold. Define how did select the 10 cases with Calcitonin levels measurement from the FNA sample.
3. Results
You described 11 cases of CMT in Group 1, and 30 cases of CMT in group 3, with 2 FNAC results that were not comprised in the final analysis – 39 cases not 41. Then you describe the Calcitonin levels, that were measured in some of the cases (as is suggested by your text) Were all 41 Confirmed cases analysed with Calcitonin? You describe 40/41 cases with increased Calcitonin levels—what happened with the case nr 41 ? Indirectly from the table the reader can understand that Calcitonin levels were normal in the 1/41 cases. Explain how come all CMT cases had Calcitonin levels measurements but not all evaluated cases in the study? Were there any selection biases affecting the procedure?
You are comparing the sets of data – Sensitivity and specificity are affected also by the prevalence of the disease in the study group – you have 41 CMT cases / 555 – in the mRNA group, respectively 41/384 Calcitonin group…the same is valid for PPN and NPV…. And the selection criteria for Calcitonin universal evaluation was not presented …. And if is was no criteria …. It is really hard to believe that, by change, all 41 CMT cases had Calcitonin evaluation, just by chance.
Author Response
2.1 CLINICAL MATERIAL
The definition of group 1 – 11 cases with known CMT – confirmed by pathology report (after surgical removal) – retrospective analysis – it is not clear in pathology reports of FNA reports were analysed. It is indirectly suggested that from the total of 525 cases, 375 had FNAB specimens, which were analysed for the microRNA mutation, 11/375 being with definitive CMT diagnostic.
-Unfortunately, we failed to fully understand the essence of this comment. However, since the description of patient group 1 has raised questions, we have changed it:
-A total of 555 patients were enrolled in the study. The data on 525 patients (group 1 of patients) were taken from our previous studies performed to evaluate the diagnostic accuracy of the Thyroid-INFO test [25, 26, 27]. In this group, in previous studies, the expression level of miRNA-375 was determined for all the samples, and for some samples (n = 354) the basal calcitonin level was known. In these studies, the criteria for inclusion in the study were as follows: the patients' age being at least 18 years, the presence of primary cytological slides with a sufficient amount of cellular material (Bethesda category II-VI), and postoperative histological conclusion being known. Expression of miRNA-375 was determined in the material washed off from cytological slides.
The definition of group 2 – 30 cases of CMT patients receiving treatment at the .... what do you mean under this statement? How was the definitive diagnostic for CMT made – FNAB only or surgery – if it was surgery why are these cases not comprised in the first group? What is different in group 2 versus group1 ... if they have no final pathology report confirming CMT 0 what was the golden standard for CMT diagnostic ...
This paragraph needs clarification.
-In this case, group 2 involves the new samples (not described earlier) that were analyzed in 2022. Group 1 refers to the samples already described in our previous studies, but we did not compared miRNA-375 and calcitonin in them. For all the samples (groups 1 and 2), final diagnosis was made by postoperative histological examination. We have added the following fragment to the manuscript:
-The final diagnosis for this group of patients, like for group 1, was made using the postoperative histological report. Histological examination was carried out by pathologists from the corresponding institution.
-We understand the term “receiving treatment” to mean “subjected to preoperative examination and subsequent surgical treatment”.
-In addition, a scheme for analyzing microRNA-375 and preoperative basal calcitonin in group 1 and 2 patients was added.
2.2 MOLECULAR ANALYSIS
Group1 - mutation were part of a more comprehensive molecular testing.
-Under our previous studies, we have identified a larger number of markers for group 1 patients (not solely expression of miRNA-375); however, only mutation in the BRAF gene has been detected, no mutations in the RET gene were revealed.
Group 2 – define what you understand under” de novo” – did you test miRNA 375 expression only?
-The term “de novo” is probably not a good one. We meant that the miRNA-375 level was measured only in this study, but not earlier. Yes, we did not test other markers within the Thyroid-INFO test under this study. The sentence has been changed in the manuscript:
-In the specimens from group 2 patients, the miRNA-375 levels were measured under this study.
2.3 Calcitonin
Defined values of Calcitonin, threshold. Define how did select the 10 cases with Calcitonin levels measurement from the FNA sample.
-The cut-off values for calcitonin (regardless of sex) were as follows: < 10 pg/mL – normal value; 10–100 pg/mL – the “gray zone”; and > 100 pg/mL – medullary thyroid cancer.
-As for the 10 cases, those were the patients for whom biopsy samples were taken not only from the node, but also from the ipsi-/contralateral thyroid lobe (as reported in the Results section); calcitonin was also detected in the wash-out fluid from fine-needle aspiration of thyroid nodules. All these patients were from group 2, but their number was not 41, since not all the patients agreed to undergo an additional biopsy.
- Results
You described 11 cases of CMT in Group 1, and 30 cases of CMT in group 3, with 2 FNAC results that were not comprised in the final analysis – 39 cases not 41. Then you describe the Calcitonin levels, that were measured in some of the cases (as is suggested by your text) Were all 41 Confirmed cases analysed with Calcitonin? You describe 40/41 cases with increased Calcitonin levels—what happened with the case nr 41 ? Indirectly from the table the reader can understand that Calcitonin levels were normal in the 1/41 cases. Explain how come all CMT cases had Calcitonin levels measurements but not all evaluated cases in the study? Were there any selection biases affecting the procedure?
- We have assessed the miRNA level in preoperative cytological slides. In 2 out of 41 cases, the cytological conclusion was non-informative, so these patients were removed from consideration.
- The preoperative blood calcitonin level was known in all 41 patients with MTC.
- In one patient with MTC, the preoperative calcitonin level was < 1.0 pg/mL for an unknown reason. This patient has undergone surgical treatment, like all the 555 patients in our study.
- In Russia, calcitonin level measurement is an obligatory procedure before any surgery for nodular thyroid pathology. It is most likely that the calcitonin level was known in all 555 patients, but we did not have these data in some group 1 patients (n = 171). The difficulties were related to the fact that there was no access to all the archived medical records, since the patients had been undergoing treatment at different study centers in Russia. In the previous studies involving these patients, calcitonin level was not a parameter we were interested in.
You are comparing the sets of data – Sensitivity and specificity are affected also by the prevalence of the disease in the study group – you have 41 CMT cases / 555 – in the mRNA group, respectively 41/384 Calcitonin group…the same is valid for PPN and NPV…. And the selection criteria for Calcitonin universal evaluation was not presented …. And if is was no criteria …. It is really hard to believe that, by change, all 41 CMT cases had Calcitonin evaluation, just by chance.
-The fact is that prevalence of the disease does not influence sensitivity and specificity; it affects only PPN and NPV. It has been mentioned in the Discussion section.
-In our study, for a sample where MTC was 10% of all samples, the PPV was 80%. However although sensitivity and specificity characterize a test independently of the incidence of cancer, but NPV and PPV depend on the incidence. Assuming fixed sensitivity and specificity, Bayes’ theorem can predict the dependence of a test’s NPV and PPV on the incidence of cancer, so with the incidence of MTC among all thyroid nodules (0.25%), the PPV will also be 7.7% (95% CI: 4.1–13.4%).
-As for the calcitonin measurement criteria, this question has already been answered: the data for calcitonin were available for all the samples, but we were unable to obtain them for some sample, since in the initial studies involving these patients, we were not interested in calcitonin level. For the case of medullary thyroid cancer samples, calcitonin level was a parameter we were interested in, so we have these data.
Reviewer 4 Report
Thank the Editor to give me the opportunity to revise this article. The manuscript is of great interest in the field of current research. I read this article with great interest. However there are several points which need to be revised and improved. Fr example:
1) I suggest authors to shorten the "Introduction", moving some elements from this section to the "Discussion"(ie the comparison with previous studies)
2)Line 60-62: please mention also the drugs (such as B-blockers..) that can cause high levels of calcitonin
3)I suggest authors to revise the "Methods" section which is not clear (gfor example group 2 is not specified enough)and it needed to be more complete and coincise
4) A "conclusion" section is lacking
Author Response
1) I suggest authors to shorten the "Introduction", moving some elements from this section to the "Discussion"(ie the comparison with previous studies).
We have no impression that the Introduction section involves text that can be moved to the Discussion section; therefore, the fragment pertaining to previous studies has been deleted from the Introduction.
2) Line 60-62: please mention also the drugs (such as B-blockers..) that can cause high levels of calcitonin.
Thank you for the comment, we have added the following text:
“…and also by pharmacological agents (i.e., proton-pump inhibitors, glucocorticoids, and β-blockers).”
3) I suggest authors to revise the "Methods" section which is not clear (for example group 2 is not specified enough) and it needed to be more complete and coincise.
We have changed the description of sample groups to some extent:
A total of 555 patients were enrolled in the study. The data on 525 patients (group 1 of patients) were taken from our previous studies performed to evaluate the diagnostic accuracy of the Thyroid-INFO test [25, 26, 27]. In this group, in previous studies, the expression level of miRNA-375 was determined for all the samples, and for some samples (n = 354) the basal calcitonin level was known. In these studies, the criteria for inclusion in the study were as follows: the patients' age being at least 18 years, the presence of primary cytological slides with a sufficient amount of cellular material (Bethesda category II-VI), and postoperative histological conclusion being known. Expression of miRNA-375 was determined in the material washed off from cytological slides.
Since the percentage of MTC in group 1 patients is low, cytology specimens obtained by FNAB were additionally examined in 30 patients with MTC (control group 2) who were receiving treatment at the National Medical Research Center for Endocrinology. The final diagnosis for this group of patients, like for group 1, was made using the postoperative histological report. Histological examination was carried out by pathologists from the corresponding institution. The miRNA-375 level was evaluated in the material washed off from cytological slides in all the group 2 patients…
In addition, a scheme for analyzing microRNA-375 and preoperative basal calcitonin in group 1 and 2 patients was added.
4) A "conclusion" section is lacking.
We have added the Conclusion section to the manuscript:
Conclusion
Our results show that assessment of miRNA-375 expression enables high-accuracy detection of medullary thyroid carcinoma, but additional studies need to be performed because of the relatively small sample of patients.
Round 2
Reviewer 3 Report
2.1 CLinical material : The definition of group 1 -11 cases with known CMT confirmed be pathology report (after surgical removal) - retrospective analysis - it is not clear .. from the total of 525 cases *375 of them having FNAB evaluation + Micro RNA mutations, 11/375 being with definitive C;T diagnostic? Where there no cases of CMT in the other 150 cases (with no FNAB evaluation?)
The definition of group 2 - 30 cases of CMT patients receiving treatment at the..... what do you mean under this statement/ How was the definitive diagnostic made ? - with FNA; with pathology report? And if it is like this why do we have group 2? Please redefine the difference between group 1 and 2? What was the golden standard for the CMT diagnostic?
2.2 Molecular analysis
Group 1 - mutation were part of a more complete molecular testing.
Group 2 - define what do you understand under ”de novo” - did ypu teste miRNA 375 only?
2.3 CAlcitonin
Define normal Calcitonin values, threshold suggestive for CMT diagnostic used in your Department. Define how did you select the 10 cases with Calcitonin levels appraised also in the FNA specimen.
3. Results
You described 11 cases of CMT in Grou 1, and 30 cases of CMT in group 3, with 2 FNAC results that were not comprised in the final analysis = 39 cases and not 41 cases. Then you describe the Calcitonin levels, that were measured in some of the cases (as it is suggested in your text) . Where all 41 confirmed cases wt CMT, previously analysed by Calcitonin measurements? You described 40/41 cases with increased Calcitonin levels- please clarify what happend with case 41. Explain how come all CMT cases had Calcitonin levels measurements but nor all evaluated cases in the study? Where there any selection biases affecting the study?
You are comparing the sets of data - sensitivity and specificity are affe ted also by the prevalence of the disease in the study group ypu have 41 CM cases/555 cases (the microRNA group_ respectively 41/384 cases with CAlcitonin meassurement.... and if it was no selection criteria ,,,it is really surprising that there was co CMT case in the noncalcitonin evaluated group.
Author Response
I would like to draw your attention to the fact that we have already answered all these questions at the previous stage of the review. If these answers turned out to be somewhat unsatisfactory, then we would like to receive a comment on this matter so that we can correct the manuscript or our answers.
2.1 CLinical material : The definition of group 1 -11 cases with known CMT confirmed be pathology report (after surgical removal) - retrospective analysis - it is not clear .. from the total of 525 cases *375 of them having FNAB evaluation + Micro RNA mutations, 11/375 being with definitive C;T diagnostic? Where there no cases of CMT in the other 150 cases (with no FNAB evaluation?)
We have changed the description of patient group 1:
Lines 143-151:
A total of 555 patients were enrolled in the study. The data on 525 patients (group 1 of patients) were taken from our previous studies performed to evaluate the diagnostic accuracy of the Thyroid-INFO test [25, 26, 27]. In this group, in previous studies, the expression level of miRNA-375 was determined for all the samples, and for some samples (n = 354) the basal calcitonin level was known. In these studies, the criteria for inclusion in the study were as follows: patients' age being at least 18 years, the presence of primary cytological slides with a sufficient amount of cellular material (Bethesda category II-VI), and postoperative histological conclusion being known. Expression of miRNA-375 was determined in the material washed off from cytological slides (Fig. 1).
In addition, a scheme for analyzing microRNA-375 and preoperative basal calcitonin in group 1 and 2 patients was added.
The definition of group 2 - 30 cases of CMT patients receiving treatment at the..... what do you mean under this statement/ How was the definitive diagnostic made ? - with FNA; with pathology report? And if it is like this why do we have group 2? Please redefine the difference between group 1 and 2? What was the golden standard for the CMT diagnostic?
In this case, group 2 involves the new samples (not described earlier) that were analyzed in 2022. Group 1 refers to the samples already described in our previous studies, but we did not compared miRNA-375 and calcitonin in them. For all the samples (groups 1 and 2), final diagnosis was made by postoperative histological examination. We understand the term “receiving treatment” to mean “subjected to preoperative examination and subsequent surgical treatment”.
We have added the following fragment to the manuscript:
Lines 160-163:
The final diagnosis for this group of patients, like for group 1, was made using the postoperative histological report. Histological examination was carried out by pathologists from the corresponding institution.
2.2 Molecular analysis
Group 1 - mutation were part of a more complete molecular testing.
Under our previous studies, we have identified a larger number of markers for group 1 patients (not solely expression of miRNA-375); however, only mutation in the BRAF gene has been detected, no mutations in the RET gene were revealed.
Group 2 - define what do you understand under ”de novo” - did ypu teste miRNA 375 only?
The term “de novo” is probably not a good one. We meant that the miRNA-375 level was measured only in this study, but not earlier. Yes, we did not test other markers within the Thyroid-INFO test under this study. The sentence has been changed in the manuscript:
Lines 196-197: In the specimens from group 2 patients, the miRNA-375 levels were measured under this study.
2.3 CAlcitonin
Define normal Calcitonin values, threshold suggestive for CMT diagnostic used in your Department. Define how did you select the 10 cases with Calcitonin levels appraised also in the FNA specimen.
We have added the following fragment to the manuscript:
Lines 241-243:
The cut-off values for calcitonin (regardless of sex) were as follows: < 10 pg/mL – normal value; 10–100 pg/mL – the “gray zone”; and > 100 pg/mL – medullary thyroid cancer.
As for the 10 cases, those were the patients for whom biopsy samples were taken not only from the node, but also from the ipsi-/contralateral thyroid lobe (as reported in the Results section); calcitonin was also detected in the wash-out fluid from fine-needle aspiration of thyroid nodules. All these patients were from group 2, but their number was not 41, since not all the patients agreed to undergo an additional biopsy.
- Results
You described 11 cases of CMT in Grou 1, and 30 cases of CMT in group 3, with 2 FNAC results that were not comprised in the final analysis = 39 cases and not 41 cases. Then you describe the Calcitonin levels, that were measured in some of the cases (as it is suggested in your text) . Where all 41 confirmed cases wt CMT, previously analysed by Calcitonin measurements? You described 40/41 cases with increased Calcitonin levels- please clarify what happend with case 41. Explain how come all CMT cases had Calcitonin levels measurements but nor all evaluated cases in the study? Where there any selection biases affecting the study?
- We have assessed the miRNA level in preoperative cytological slides. In 2 out of 41 cases, the cytological conclusion was non-informative, so these patients were removed from consideration.
- The preoperative blood calcitonin level was known in all 41 patients with MTC.
- In one patient with MTC, the preoperative calcitonin level was < 1.0 pg/mL for an unknown reason. This patient has undergone surgical treatment, like all the 555 patients in our study.
- In Russia, calcitonin level measurement is an obligatory procedure before any surgery for nodular thyroid pathology. It is most likely that the calcitonin level was known in all 555 patients, but we did not have these data in some group 1 patients (n = 171). The difficulties were related to the fact that there was no access to all the archived medical records, since the patients had been undergoing treatment at different study centers in Russia. In the previous studies involving these patients, calcitonin level was not a parameter we were interested in.
You are comparing the sets of data - sensitivity and specificity are affe ted also by the prevalence of the disease in the study group ypu have 41 CM cases/555 cases (the microRNA group_ respectively 41/384 cases with CAlcitonin meassurement.... and if it was no selection criteria ,,,it is really surprising that there was co CMT case in the noncalcitonin evaluated group.
Prevalence of the disease does not influence sensitivity and specificity; it affects only PPV and NPV. It has been mentioned in the Discussion section.
Lines 363-368:
In our study, for a sample where MTC was 10% of all samples, the PPV was 80%. However although sensitivity and specificity characterize a test independently of the incidence of cancer, but NPV and PPV depend on the incidence. Assuming fixed sensitivity and specificity, Bayes’ theorem can predict the dependence of a test’s NPV and PPV on the incidence of cancer, so with the incidence of MTC among all thyroid nodules (0.25%), the PPV will also be 7.7% (95% CI: 4.1–13.4%).
As for the calcitonin measurement criteria, this question has already been answered: the data for calcitonin were available for all the samples, but we were unable to obtain them for some sample, since in the initial studies involving these patients, we were not interested in calcitonin level. For the case of medullary thyroid cancer samples, calcitonin level was a parameter we were interested in, so we have these data.

Reviewer 4 Report
Accept in the present form
Author Response
Thanks for your review!